# Beyond Body Weight: Design and Validation of Psycho-Behavioural Living and Eating for Health Segments (LEHS) Profiles for Social Marketing

**DOI:** 10.3390/nu12092882

**Published:** 2020-09-21

**Authors:** Linda Brennan, Shinyi Chin, Annika Molenaar, Amy M. Barklamb, Megan SC Lim, Mike Reid, Helen Truby, Eva L. Jenkins, Tracy A. McCaffrey

**Affiliations:** 1School of Media and Communication, RMIT University, Melbourne 3004, Australia; linda.brennan@rmit.edu.au (L.B.); shinyi.chin@rmit.edu.au (S.C.); 2Department of Nutrition, Dietetics and Food, Monash University, Notting Hill 3168, Australia; annika.molenaar@monash.edu (A.M.); amy.barklamb@monash.edu (A.M.B.); eva.jenkins@monash.edu (E.L.J.); 3Behaviours and Health Risks, Burnet Institute, Melbourne 3004, Australia; megan.lim@burnet.edu.au; 4Melbourne School of Population and Global Health, University of Melbourne, Carlton 3053, Australia; 5School of Economics, Finance and Marketing, RMIT University, Melbourne 3000, Australia; mike.reid@rmit.edu.au; 6School of Human Movement and Nutrition Sciences, The University of Queensland, Brisbane 4072, Australia; h.truby@uq.edu.au

**Keywords:** social marketing, social media, healthy eating, young adults, obesity prevention, instrument development

## Abstract

Obesity, sedentary behaviour, and poor dietary habits amongst young adults are growing concerns, with this age group being in a worse state of health and nutrition than adolescents and adults. This paper presents the procedures for establishing a new instrument for defining behaviours in relation to healthy lifestyle and food choices amongst young adults (Living and Eating for Health Segments: LEHS). The aim of this paper is to outline the instrument design protocol for external validation and to permit replication in other studies. The instrument design process used a multi-step social marketing instrument design method. This approach has previously been used in designing valid and reliable measures in marketing and consumer research, including social marketing. The protocol established six psycho-behavioural LEHS profiles for young adults. These profiles are: Lifestyle Mavens (15.4%), Aspirational Healthy Eaters (27.5%), Balanced-all Rounders (21.4%), Health Conscious (21.1%), Contemplating Another Day (11.2%), and Blissfully Unconcerned (3.4%). Each of these profiles provided insights into psycho-behavioural characteristics that can be used in designing apposite social media social marketing campaigns.

## 1. Introduction

Obesity, or abnormal or excessive fat accumulation, is amongst the largest public health issues in modern society, with rates of obesity having tripled since 1975 [1]. The associated risk of disease from excessive fat accumulation [2] has contributed significantly to the overall burden of disease [3]. Modern environments are obesogenic in nature, with built environments hindering active transport and physical activity opportunities and the food environment favouring highly processed convenience foods, which are often nutrient-poor and energy-dense [4]. Social, mental, and demographic factors such as income, stress, and the socioeconomic status of your neighbourhood have also been found to contribute to poor weight management [4]. Young adults (aged between 18 and 24 years) are in a particularly vulnerable stage of life as they transition from secondary education to tertiary education or the workforce [5]. This stage of life, coined “Emerging Adulthood”, comes with unique challenges related to recent social and financial independence, development of identity, and finding their place in the world [5]. The health of young adults has been found to be worse when compared to adolescents and adults [6]. Diet quality [7,8] and physical activity levels [9] have been seen to decline during this transitional stage, which may be linked to the prevalence of weight gain [10,11], and with this increasing weight trajectory, the number of individuals with obesity is rising.

Currently, the most frequently used and accepted method of measurement for obesity is body mass index (BMI, kg/m^2^), which takes into account an individual’s body weight and height. Due to the relative ease of collection of measurements of height and weight, BMI is an accessible measure that can be used on a large scale. It is a surrogate measure of body fat as BMI has been found to have low sensitivity to be able to detect individuals who have excessive body fat [12] and can lead to misclassifications of non-obese individuals [13]. Excessive body fat and regional distribution of fat deposits, particularly around the waist, are risk factors for many non-communicable diseases, not necessarily a person’s body weight [14]. The crude nature of the BMI measurement does not take into account body composition such as percentage of lean and fat mass [12], therefore it may not accurately represent someone’s risk of disease.

Importantly, while valuable as a predictor of risk, BMI is not particularly useful as a tool for informing interventions designed to reduce risks and induce the behavioural changes necessary to engender healthy lifestyles. Studies have shown that behavioural and psychographic segmentation is appropriate for tackling obesity [15,16]. However, the majority of papers on the topic continue to use physical and economic characteristics such as BMI, income, age, and education as a means to profile or segment the people in their research [17,18,19]. Furthermore, the appropriateness of using BMI has long-standing concerns [20]. Nevertheless, policymakers continue to use it to inform government policy on obesity, regardless of its known inadequacies [21]. 

Our interdisciplinary team comprises of academics from nutrition, social marketing, and health promotion and is approaching the issue of obesity-related behaviours from a social marketing perspective [22]. Social marketing is the use of commercial tools and techniques for social purposes [23]. One of the key tools of social marketing is that of market segmentation, which ensures that programs fostering change are aligned with the needs and wants of the market or target population [24]. Segmentation is the a priori grouping of individuals into relatively homogenous groups based on known similarities [25]. Segmentation also ensures that efforts to persuade are based on the communication competences of the target audience [26]. Consequently, thorough formative research into audience dynamics is essential prior to designing interventions: see, for example, [27]. 

There are a number of well-accepted methods of segmentation or profiling used in designing social marketing interventions or programs [24]. For example, there is psychographics, which is grouping by factors such as personality traits, beliefs, values, lifestyles, attitudes, and interests [28], as well as behavioural segmentation, which is the grouping of people based on their behaviours such as buying or using products [26]. These methods are often combined with demographic and geographic methods to produce a nuanced profile of the targeted group. Segmentation by demographics (such as age and gender) and geography are common methods used in the public health sector, but can be quite limiting in terms of developing an in-depth understanding of the target group [17]. Therefore, the Communicating Health project uses a psycho-behavioural approach to segmentation that comprises a combination of methods of segmentation [29]. 

The full combination of methods used in the overall Communicating Health project is described later in this paper and is summarised in the study protocol [22]. Using a combination of methods provides a more powerful and nuanced understanding of people’s behaviours within their social contexts and accounts for the complexities of their lived experiences [30,31]. The rationale for this paper is to explain how psycho-behavioural segmentation can produce meaningful groupings for social marketing and health promotion (HP) programs and to provide opportunities for replication by HP practitioners and health professionals in relation to targeting young adults. 

This paper sets out the procedure underpinning the development of a new profiling instrument that can be used to design HP programs that are soundly based on theories of behaviour change. Due to the interdisciplinary nature of this research, a glossary of terms (Appendix A; Table A1) has been created to define terms that may be uncommon, particularly in the field of nutrition. This glossary is a continuation of the terms present in the Communicating Health protocol study [22]. 

## 2. Materials and Methods 

### 2.1. Aims of the Procedure

The overarching aim of the procedure is to produce meaningful profiles of people for the purposes of designing and specifically targeting more accurate HP interventions. The aim of segmentation is to produce profiles that are:Sufficiently different to each other that they justify the development of a differing HP program.Measurable in terms of their prevalence within the population.Accessible in terms of being able to be effectively reached and addressed with specific HP programs.Substantial enough to warrant differential attention.

To this end, the Communicating Health project undertook to determine profiles in relation to young adult’s psychosocial characteristics and behaviours relating to healthy eating and accessing healthy eating content through social media. These “*Living and Eating for Health Segments*” (LEHS) profiles will then be used throughout subsequent phases of the Communicating Health study to determine their validity in a wider audience and to identify effective ways to communicate, motivate, and engage with young adults from different segments through social media [22].

### 2.2. Instrument Development Procedure

A key assumption underpinning the development of measurable profiles is that the concept is measurable in the first instance [32] and therefore those core elements are both observable and reportable (i.e., manifest). In order to ensure that the embedded concepts were identified, a multi-stage, mixed-methods approach was adopted (See Figure 1).

One of the guiding principles of the adopted approach was ensuring the validity of the LEHS profiles [33]. Brennan and colleagues (2011) proposed a taxonomy of types of validity using two axes. The first axes consist of whether a “measure” or “method” is being validated. In most studies, validation efforts often focus on achieving a valid “measurement” (e.g., constructs that are reliable and reproducible). However, it is important to also consider the validity of the overall “method”—the research process from input to outcome. The second axes determine if the validity is “formative” or “prognosticative”. The process of formative validity takes place before data collection, while prognostication validity aims to understand cause and effect tests. As such, different types of validity tests can be categorised into the following combinations:Formative-method: essential during the initial stages of research to establish what a construct/idea/concept is or is not (including definition and its defining characteristics).Formative-measure: to test whether the “real world” observations captured the abstract concept as defined in the previous stage.Prognosticative-method: to ensure the research process is both rigorous and consistent (where required).Prognosticative-measure: to establish whether a measure behaves in a way that it was expected to in relation to other constructs in a theory [33].

The different types of validity and how they relate to the stages of our research approach are outlined in Table 1 and are further explained in the following paragraphs.

### 2.3. Literature Reviews and Formative Research

The Communicating Health Project inception was in August of 2016. Since then, three scoping and literature reviews and baseline qualitative research articles have contributed to the conceptualisation of the LEHS profiles.

#### 2.3.1. Literature Reviews

The literature reviews conducted prior to the qualitative analysis of the online conversations shaped the context of the food and health environment in which the young adults in the online conversations were living and this was used as a lens while analysing them.
A systematic review into social media use for nutrition in young adults was undertaken [34]. This review found that social media is an acceptable platform to disseminate information about healthy eating and recipes by young adults. However, social media was generally included only as one aspect of a complex intervention. Interventions as a whole (not just the social media component) had a positive statistically significant impact on nutritional outcomes in 1/9 trials. Reasons for low engagement with social media included the use of post types that are not interactive and being asked to talk about personal weight/weight loss on an open social media platform.In order to understand the perspectives of Indigenous Australians, a scoping review was undertaken [35]. The aim of this study was to examine the extent of health initiatives using social media that aimed to improve the health of Australian Aboriginal communities.A systematic review of the impact of social media on body image and nutrition found that [36] social media health-related content should refrain from focusing on body weight or physical appearance as measures of health because they are likely to alienate young adults rather than encourage behaviour change.

#### 2.3.2. Formative Research

The formative research from the online conversations was conducted additionally to the LEHS thematic analysis in order to explore in depth specific topics within the online conversations. This formative research helped to inform and validate the LEHS profiles.
A further study [37] demonstrated that social media strategies applied by influencers attract a large audience and engagement. Furthermore, HP professionals’ messages are less effective than celebrity influencers. The study found that social media, particularly Instagram, facilitates para-social interactions where imaginary social relationships and interpersonal interactions between the lifestyle personality and the social media user occur. Participants who experience positive emotions when viewing a post on social media are far more likely to engage with that post than those who do not experience positive emotions.Baseline exploration of aspects of the online conversations related to the language of health [38] found that young adults had a holistic view of health and that competing demands hindered their ability to realise healthy behaviours. Current healthy eating messaging did not address their needs.Analysis of the qualitative research [39] identified that consumer segmentation and social marketing techniques can assist health professionals to understand their target audience and tailor specific messages to different segments. Psycho-behavioural segmentation also provides unique insights on which groups may be most easily influenced to adopt the desired behaviours.Participants described how social media influenced their decisions to change their health behaviours [40]. Access to social support and health information through online communities were juxtaposed with exposure to highly persuasive fast-food advertising. Some participants expressed that exposure to online health content induced feelings of guilt about their behaviour, which was more prominent among females. Poor health behaviours associated with social activities and fast-food advertising were discussed as major barriers to change.

At this stage, both formative and prognosticative validity were present as initial literature and systematic reviews were conducted to answer the epistemological question of “what is the nature of knowledge” in this area. Here, semantic validity helped determine if there is a uniform semantic usage or not. Nomological validity established any theoretically supported construct relationships from prior research.

### 2.4. Online Conversations with Young Adults

Data were collected from 195 participants who completed the four-week online conversations, the formative Phase 1a of the Communicating Health study [22]. These online conversations sought to gather information about health and well-being, especially in relation to food in an informal online social setting. 

The research team aimed to recruit 200 young adults aged 18–24 years old for the online conversations. The recruitment target was set to achieve an extensive amount of information and was based on previous research that utilised a similar methodology to the online conversations [41]. Participants were recruited by an Australian Market and Research Society-certified [42] market research field house. Three online research panels were utilized, which consisted of people who had voluntarily signed up in the expectation of being invited to complete surveys and studies. Three panels were used to ensure a wide mix across the sample as well as to reach the target quotas. Quotas were set to be approximately representative of the Australian population [43] in terms of location (both Australian State or Territory and location type i.e., metropolitan and regional locations) and gender. These panels were accredited for the purpose of market research by the International Organization for Standardization (ISO) [44]. 

Panel members were sent a screening survey to determine their eligibility [45]. Panel members were eligible if they were aged between 18–24 years old, self-reported using social media at least twice a day, and were currently residing in Australia. Completers of this survey who were eligible (*n*234) were sent a link to complete a profiling survey and to the register on the online conversations website. Once registered, they were assigned into one of four separate online communities based on their age (18–21 years or 22–24 years) and their interest in health (low or mid/high). Interest in health was classified based on the median value from the following question in the screening survey “On a scale of 1–7 where 1 means “Strongly Disagree” and 7 means “Strongly Agree”, please indicate how strongly you agree with the following statement—“I take an active interest in my health”.

Participants within these online communities were posed questions about different topics in 20 different forums, two challenges, three short polls, and an ongoing journal entry by market research moderators (see [45] for details of activities). These activities were released at different times but remained open for participants to complete for four weeks (10th May to 6th of June 2017). Not all participants who registered completed the activities in the online conversations (Figure 2). A referral system was put in place to recruit more participants whereby participants could invite their friends who were then screened and profiled in the same way. Participants received an AU$100 voucher for completing all activities and the 20 most active participants (five from each community) received an extra AU$100 voucher. Figure 2 summarises the stages in participant recruitment. 

From these online conversations, observational and face validity ensured concepts were reducible to observations and that concepts “looks as if” they should measure particular attributes. At this stage, formative validity for both measure and method were used ontologically (what is there that can be known?) and methodologically (how can the researcher go about finding out whatever can be known?).

The participant profile for the online conversations is presented in Table 2.

### 2.5. Qualitative Thematic Analysis

Preliminary profile development was undertaken in a multi-stage process involving an interdisciplinary team (K.K., L.B., M.R.) including the market research agency involved in collecting the data. The agency team included the moderator of the groups (J.K.), a consumer psychologist (D.G.), and an anthropologist (M.K.). Draft profiles were developed based on thematic analysis of the online conversations data and background literature reviews (K.K., L.B., M.K.). Qualitative data analysis used two different approaches: hypothesis-generating (to uncover new themes not previously identified by literature reviews) and hypothesis-testing to identify statements relating to the Integrated Model of Behaviour Change (IMBC) model [46]. For the hypothesis-generating analysis, the online conversations were analysed in an exploratory way using a hand-coding process on paper and using NVivo qualitative data analysis Software; QSR International Pty Ltd. Version 11 (Melbourne, Australia). Researchers used thematic analysis to examine the qualitative data collected from participants’ discussions and identify themes using a constant comparison approach [47]. Common themes associated with healthy eating behaviour, using the IMBC model components, were drawn out and used as a basis for considering likely LEHS profiles. Additionally, comparative analysis was used on a continual basis to compare people and each group to check that LEHS profiles are appropriately represented by the data. Investigator triangulation enhanced the transferability and dependability of the research findings [48].
**Profiles reviewed independently by research team members:**Profiles were reviewed independently by all team members (K.K., L.B., M.R., S.C., T.A.M., M.S.L, H.T., A.M., E.J.) and disagreements were resolved via consensus in a series of single issue focus meetings.At this stage, semantic validity determined if there were uniform semantic usages for the profiles identified from the online conversations. The purpose of this formative-method validation is methodological.**Expert panel review of profiles:**Profiles were iterated based on this feedback cycle and summaries were developed that could be used in online data collection procedures (K.K., L.B.).Summaries were evaluated by the whole team before being tested with a sample cohort of young adults (Honours students enrolled in programs at Royal Melbourne Institute of Technology (RMIT) and Monash University as well as two from the University of Ulster who were on placement in Australia at the time).Following the previous research stage, iterated profiles were further validated (prognosticative-measure). Content validity determined the degree to which the profiles can be generalised. Here, validity helped answer methodological and axiological (what is intrinsically worthwhile?) questions.**Think Tank review and sense-check of profiles:**Subsequent to the development of the LEHS, a Think Tank was held with the research team and partner organisations to review the findings of the online conversations and validation survey.The LEHS profiles were sense checked and further defined via iteration with team members and Think Tank participants. Potential ideas for evidence-based HP campaigns targeting the different LEHS and their different attitudes, behaviours, and needs were also discussed.This Think Tank was also used to inform further stages of the Communicating Health study, which involved the co-creation of HP campaigns with young adults [22].At this stage, the LEHS were then further validated to ensure that the operationalisation measures the profiles as it purports to measure (construct validity). The purpose of this formative-measure validation is an epistemological one.

### 2.6. Online Survey Testing LEHS

Following multiple iterations and development of the profiles on a feedback cycle by the research team members, an expert panel, and Think Tank, the outcome LEHS profiles were then quantitatively defined using an online survey methodology. An online survey was conducted in December 2018 with *n*2019 young adults aged 18 to 24 years old residing in Australia. The survey consisted of 46 closed-ended questions and self-reported height and weight. Questions included demographics, quality of life, nutrition knowledge, food and cooking skills, social media use, and classification into one of the six LEHS profiles. The extended methods and results for the detailed analysis of the LEHS from this survey will be published in an upcoming manuscript. Differences between demographics in the LEHS profiles outlined in this paper were determined using IBM SPSS statistics^®^ version 25 (Armonk, NY, USA). One-way ANOVA with post-hoc testing for age and BMI was conducted. Tests were adjusted for all pairwise comparisons using the Bonferroni correction. Pearson’s chi-square test was performed for gender, studying, and income.

Through this online survey, the LEHS were further validated on both a methodological and an ontological level. Construct and nomological validity were conducted to ensure that the operationalisation of the LEHS worked as a measure and to determine interrelationships with other related constructs. The results of this will be reported elsewhere. 

## 3. Results

The LEHS profiles that were created as a result of this extensive process are outlined in Table 3.

The LEHS profiles were examined for basic differences using the following demographics (Table 4) in a sample of *n*2019 young adults who completed the online survey. There were statistically significant (*p* < 0.05) differences between the LEHS profiles for gender, BMI, educational status, and weekly income. 

There were no significant differences between LEHS based on age (Table 4). Considering that the survey was targeted to a narrow age range—young adults, this is to be expected. 

There were statistically significant differences between LEHS based on gender, with males indicating that they are more likely to categorise themselves as “Health Conscious” (53.6%) than females of the same age (43.5%) and “Lifestyle Mavens” males (62.1%) and females (36.0%). On the other hand, the “Aspirational Healthy Eaters” category included more females (61%) than males (35.4%). This imbalance extended into the “Balanced All Rounder” group, with males at 34.7% and females at 62%. These results potentially illustrate an unwillingness to “lead” the way by advocating for healthy eating. The LEHS profile “Contemplating Another Day” also included significantly more females (51.8%) than males (43.8%). However, the “Blissfully Unconcerned” included more males (56.5%) than females (36.2%), indicating that males might be caught up in an “all or nothing” internal debate—having to be a super healthy maven or not bothering at all. 

While there were significant differences between the LEHS when it came to BMI, their similarities were not practically significant for designing social marketing strategies. For example, Lifestyle Mavens (24.6 kg/m^2^), the Health Conscious (23.4 kg/m^2^), and Balanced All Rounders (23.7 kg/m^2^) all reported mean BMIs within a similar range, despite having quite different attitudes and behaviours towards eating and lifestyles associated with food intake. Additionally, the Aspirational Healthy Eaters (26.0 kg/m^2^) and the Blissfully Unconcerned (26.3 kg/m^2^) reported similar BMIs. Consequently, BMI is not a useful measure for differentiating between these psycho-behavioural LEHS profiles. 

Another often used demographic segmentation method is that of education. The data show that study status was not significantly associated with any of the LEHS profiles. Educational status does not therefore appear to have an impact on these psycho-behavioural LEHS profiles. 

On the other hand, income does have an impact on these LEHS profiles, with a significant proportion of the sample being on a relatively low income (less than $400 per week). Despite this, the “Blissfully Unconcerned” were more likely than other LEHS profiles to report a lower income (43.4%). Lifestyle Mavens (15.1%) were more likely to report higher income than other LEHS, demonstrating that perhaps it requires cash to support an intensive focus on a healthy lifestyle. 

Differences in psychographics from the Communicating Health Phase 1b survey will be presented in an upcoming manuscript.

## 4. Discussion

This paper outlines the procedure underpinning the development of a new LEHS profiling instrument that can be used to design HP programs that are soundly based on theories of behaviour change. The LEHS profiling instrument was iteratively developed utilising a multi-step social marketing instrument design method including qualitative analysis and expert review. An online survey was then used to explore these LEHS profiles in a larger cohort of Australian young adults. Our results show that there are significant differences in demographics between LEHS profiles that would be helpful for people designing social marketing and HP strategies targeting specific groups. These differences in demographics between groups may have shaped the differing attitudes and values that distinguish the LEHS and adds to the defining qualities of the profiles. We encourage continued collection of such data alongside the use of the LEHS as we recognise that multiple factors determine attitudes and values.

Differences in income between the LEHS were apparent in the online survey. The “Blissfully Unconcerned” had a high percentage of people with low incomes, which may indicate that they possibly cannot afford to be concerned about eating healthily. They may have other concerns that are more pertinent than what they eat or not believe they have the choice financially to consume healthy foods, therefore potentially affecting their overall indifferent attitude towards healthy eating. Although, of course, it is not known which of these comes first (the money or the choice). Our previous research has found that many young adults have limited financial resources and sometimes limited knowledge of budgeting their finances, so healthy eating is not always at the top of their priorities [38]. Scarcity of money and time have a negative impact on both physical activity and eating behaviours, including consumption of fruit and vegetables and energy-dense, nutrient-poor foods [49].

Eating behaviours are highly complex, influenced by both internal factors and external environmental factors [50,51]. Due to the complexity of eating behaviours, it is important to take into account different individual’s specific barriers and enablers to healthy eating [52]. Nutrition HP campaigns and interventions often choose to target audiences based on demographics such as age, gender, and location, which may be due to time and budget constraints. However, a one-size-fits-all approach to nutrition behaviour change is unlikely to be efficacious as it assumes that individuals will respond similarly to the same intervention. This is evident in a recent study, which evaluated the efficacy of a web-based nutrition intervention in young adults where no significant change in diet quality was observed, despite the intervention being rated highly in terms of acceptability by participants [53]. Health focused messaging utilising demographic segmentation of target audience only, have in the past, failed to gain and maintain the attention of a young adult population [17].

The widespread use of BMI as a segmentation tool is insufficiently nuanced when it comes to designing HP campaigns designed to facilitate behaviour change. Additionally, the focus on BMI may increase obesity stigma, with personal blame being put on the individual to control their body weight [54]. Despite the view that the social influence of weight stigma may encourage health behaviour change and subsequent weight loss, it is in fact unlikely to be a beneficial tool for public health and the reduction of obesity [54]. Such obesity stigma instead has the potential to increase the health disparities people with obesity face [54,55].

Segmentation allows for a more nuanced understanding of behaviours and the beliefs and attitudes that shape those behaviours that can be targeted to encourage behaviour change. Previous research has highlighted the differences in food intake amongst consumer segments based on impulsivity and level of involvement in food preparation, with those in the “uninvolved” segment having lower intakes of vegetables and being less likely to cook from scratch than other segments [56]. Similarly, different psycho-behavioural segments of people have been shown to respond differently to social marketing programs in relation to alcohol consumption [57,58]. When it comes to tackling the complex issue of obesity, there is a need for a deeper understanding of target audience behaviours [52]. The LEHS provide the opportunity to move beyond BMI and help people move into healthier lifestyles by encouraging behaviours that relate to their lived experience and their aspirations.

Furthermore, interventions need to have an underpinning of behaviour change models or theories in order to encourage successful behaviour change [16,59]. Social marketing and behaviour change theory suggests that effective marketing starts with knowing what the “customer” wants and then solving their problem [26]. Our research shows that the different LEHS have quite different needs, wants, and motivations when it comes to their engagement with healthy lifestyles. Thus, this method of segmentation is a step forward when it comes to designing effective interventions aimed at reducing the risks associated with unhealthy lifestyles. As outlined in the Communicating Health protocol paper, the next phase of our research involves collaborating with young adults (18–24 years old) in co-designing and co-creating strategies that target each LEHS [22]. This phase is currently ongoing and in the data collection phase.

These new LEHS provide an opportunity to develop evidence-based strategies located in psycho-behavioural theories of behaviour change and enable more efficient and targeted use of resources. Interventions founded upon the LEHS can be designed for each profile as they are behaviourally different, as well as psychographically different, and will therefore respond to different strategies. This will enable nuanced and directed communication by health promoters and will facilitate a move away from “broad brush” general nutrition HP efforts aimed at the population level or segmented only by age or gender. Furthermore, we believe that these LEHS would also benefit other fields, in particular when examining other aspects of living a healthy lifestyle such as fitness/physical exercise and mental health management.

The complexity of eating behaviours and the interplay between culture, taste preference, religion, and many other factors makes food choices highly individualistic. People do not make food choices based on just one factor such as the health outcomes of food. In this study, we did not look at how religion and culture shape the designation into different LEHS groups and this warrants investigation in the future. As these LEHS profiles were based on and tested in a sample of Australian young adults, it is unclear whether the same profiles would exist in different population groups. Further use of the profiles in a different population would therefore require validation to ascertain the applicability of these LEHS profiles. The data collected did not allow for measurement of socio-economic position beyond income and education status and therefore, future research would benefit from the inclusion of a measure of socio-economic status.

Finally, it is our contention that behavioural measures are necessary for examining the potential for solving behavioural problems. Attempting to change an entire system of socio-economic dynamics (e.g., age, education, and income) is, in our opinion, out of scope for a HP strategy. It would take global collaboration and overcoming food poverty if those dynamics were to be addressed. While these remain primary issues to be addressed in tackling obesity, social marketing and HP can only play a very limited role in changing behaviour. 

## Figures and Tables

**Figure 1 nutrients-12-02882-f001:**
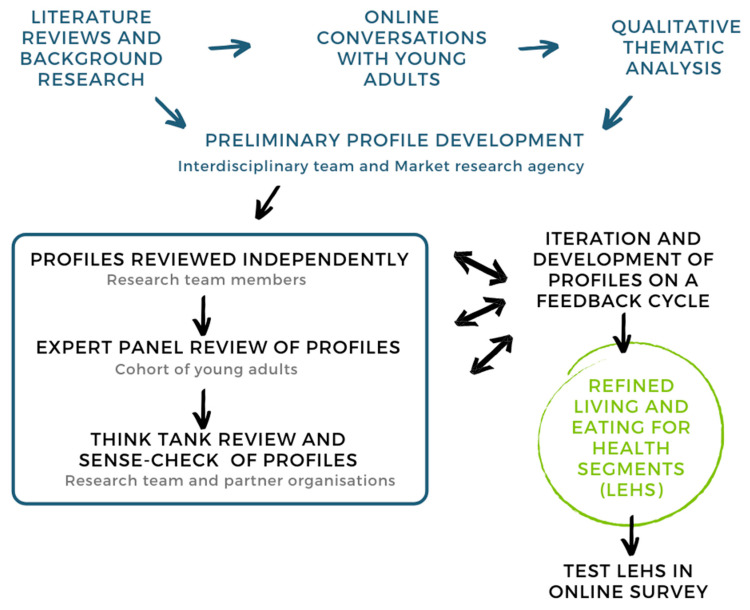
Mixed-methods approach to instrument development for Living and Eating for Health Segments.

**Figure 2 nutrients-12-02882-f002:**
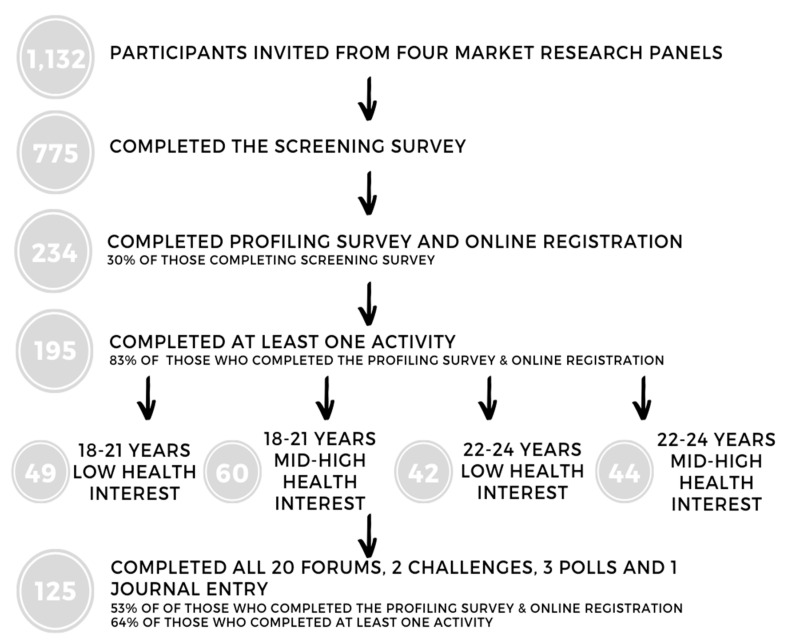
Online Conversations Participant flow chart.

**Table 1 nutrients-12-02882-t001:** Types of validity and formative research stages in LEHS profile development.

Research	Validity	Purpose
Literature reviews and background research	Semantic (formative-method) and nomological (prognosticative-measure)	Epistemology
Online conversations with young adults and subsequent qualitative analysis	Observational (formative-measure) and face (formative-method)	Methodology and ontology
LEHS profiles reviewed independently	Semantic (formative-method)	Methodology
Expert panel review of LEHS profiles	Content (prognosticative-measure)	Axiology and methodology
Think tank review and sense-check of LEHS profiles	Construct (formative-measure)	Epistemology
Online survey testing LEHS	Construct (formative-measure) and nomological (prognosticative-measure)	Methodology and ontology

LEHS: Living and Eating for Health Segments.

**Table 2 nutrients-12-02882-t002:** Participant characteristics of the online conversations (*n* = 195).

Characteristics	Categories	N (%) or Median (25th, 75th Percentiles)
Age (years)	18–21 years old	109 (56%)
	22–24 years old	86 (44%)
Gender identity ^1^	Female	119 (61%)
	Male	75 (39%)
	Non-binary/genderfluid/genderqueer	1 (1%)
Body Mass Index (BMI, kg/m^2^) categories (N = 194) ^2^	Underweight (BMI < 18.5)	16 (8%)
	Healthy weight (BMI 18.5–24.9)	106 (55%)
	Overweight (BMI 25.0–29.9)	42 (22%)
	Obese (BMI ≥ 30.0)	30 (16%)
Living location	Metro	156 (80%)
	Regional/rural	39 (20%)
Language other than English spoken at home/with parents	Yes	52 (27%)
	No	143 (73%)
Currently studying	Yes	137 (70%)
	No	58 (30%)
Level of current study ^3^	High school, year 12	8 (6%)
	TAFE, college, or diploma	18 (13%)
	University (undergraduate course)	97 (71%)
	University (postgraduate course)	14 (10%)
Highest level of completed education ^4^	High school, year 10 or lower	2 (3%)
	High school, year 11	2 (3%)
	High school, year 12	13 (22%)
	TAFE, college, or diploma	23 (40%)
	University (undergraduate degree)	16 (28%)
	University (postgraduate degree)	2 (3%)
Living arrangements ^5^	Alone	24 (10%)
	With their child(ren)	18 (8%)
	With partner	37 (16%)
	With other family	20 (9%)
	With friend(s)/housemate(s)	34 (15%)
	Living with parents	97 (42%)
Dispensable weekly income	Less than AU$40	76 (39%)
	AU$40–$79	59 (30%)
	AU$80–$119	30 (15%)
	AU$120–$199	17 (9%)
	AU$200–$299	9 (5%)
	AU$300 or over	3 (2%)
	I don’t wish to say	1 (1%)
Social media use frequency	More than twice a day	173 (89%)
	Twice a day	22 (11%)
Using social media to learn or talk about your health	Yes	128 (66%)
	No	67 (34%)
Interest in health	On a scale of 1–7, where 1 means “Strongly disagree” and 7 means “Strongly agree”, please indicate how strongly you agree with the following statement-I take an active interest in my health	6 (5, 6)
	Low interest in health (Below 6)	91 (47%)
	Mid/high interest in health (Above 6)	104 (53%)

BMI: Body Mass Index; TAFE: Technical and Further Education; ^1^ Based on the following question: “Please confirm your gender. Response options: Male; Female; Transmale/transman; Transfemale/transwoman; Non-binary/genderfluid/genderqueer; My gender is not listed (please specify)” [45]; ^2^ BMI categories based on self-reported weight and height; one participant did not answer; ^3^ Only participants currently studying answered this question; ^4^ Only participants who were no longer studying answered this question; ^5^ Participants could select more than one answer.

**Table 3 nutrients-12-02882-t003:** Living and Eating for Health Segments (LEHS) Descriptions.

Living and Eating for Health Segment	Profile Descriptions
Lifestyle Mavens	I’m passionate about healthy eating and health plays a big part in my life. I use social media to follow active lifestyle personalities or get new recipes/exercise ideas. I may even buy superfoods or follow a particular type of diet. I like to think I am super healthy.
Health Conscious	I’m health-conscious and being healthy and eating healthy is important to me. Although health means different things to different people, I make conscious lifestyle decisions about eating based on what I believe healthy means. I look for new recipes and healthy eating information on social media.
Aspirational Healthy Eaters	I aspire to be healthy (but struggle sometimes). Healthy eating is hard work! I’ve tried to improve my diet, but always find things that make it difficult to stick with the changes. Sometimes I notice recipe ideas or healthy eating hacks, and if it seems easy enough, I’ll give it a go.
Balanced All Rounders	I try and live a balanced lifestyle, and I think that all foods are okay in moderation. I shouldn’t have to feel guilty about eating a piece of cake now and again. I get all sorts of inspiration from social media like finding out about new restaurants, fun recipes and sometimes healthy eating tips.
Contemplating Another Day	I’m contemplating healthy eating but it’s not a priority for me right now. I know the basics about what it means to be healthy, but it doesn’t seem relevant to me right now. I have taken a few steps to be healthier but I am not motivated to make it a high priority because I have too many other things going on in my life.
Blissfully Unconcerned	I’m not bothered about healthy eating. I don’t really see the point and I don’t think about it. I don’t really notice healthy eating tips or recipes and I don’t care what I eat.

**Table 4 nutrients-12-02882-t004:** Participant characteristics of online survey assessing Living and Eating for Health Segments (*n* = 2019).

Characteristic	Category	Lifestyle Mavens *n*311 (15.4%)	Health Conscious *n*425 (21.1%)	Aspirational Healthy Eaters *n*556 (27.5%)	Balanced All Rounders *n*432 (21.4%)	Contemplating Another Day *n*226 (11.2%)	Blissfully Unconcerned *n*69 (3.4%)	*p* Value ^1^
Age		21 (2) ^2^	21 (2)	21 (2)	21 (2)	21 (2)	20 (2)	0.103
Gender	Male	193 (62.1%) ^3^	228 (53.6%)	197 (35.4%)	150 (34.7%)	99 (43.8%)	39 (56.5%)	<0.001
	Female	112 (36.0%)	185 (43.5%)	339 (61.0%)	268 (62.0%)	117 (51.8%)	25 (36.2%)	
	Non-binary/genderfluid/genderqueer/transgender	5 (1.6%)	11 (2.6%)	19 (3.4%)	14 (3.2%)	9 (4.0%)	4 (5.8%)	
	Prefer not to say	1 (0.3%)	1 (0.2%)	1 (0.002%)	0 (0%)	1 (0.004%)	1 (1.4%)	
Body Mass Index (kg/m^2^)		24.6 (5.9) _a,d,e_	23.4 (4.9) _a_	26.0 (6.7) _c_	23.7 (4.9) _a,b_	25.4 (6.3) _c,d_	26.3 (7.3) _b,c,e_	<0.001
	Underweight (BMI < 18.5)	28 (9.0%)	42 (9.9%)	37 (6.7%)	41 (9.5%)	16 (7.1%)	9 (13.0%)	
	Healthy weight (BMI 18.5–24.9)	171 (55.0%)	275 (64.7%)	260 (46.8%)	254 (58.8%)	111 (49.1%)	30 (43.5%)	
	Overweight (BMI 25.0–29.9)	72 (23.2%)	76 (17.9%)	145 (26.1%)	87 (20.1%)	53 (23.5%)	13 (18.8%)	
	Obese (BMI ≥ 30.0)	40 (12.9%)	32 (7.5%)	114 (20.5%)	50 (11.6%)	46 (20.4%)	17 (24.6%)	
Currently studying	Yes	171 (55.0%)	237 (55.8%)	297 (53.4%)	238 (55.1%)	132 (58.4%)	26 (37.7%)	0.016
	No	129 (41.5%)	180 (42.4%)	248 (44.6%)	185 (42.8%)	86 (38.1%)	37 (53.6%)	
	Prefer not to say	11 (3.5%)	8 (1.9%)	11 (2.0%)	9 (2.1%)	8 (3.5%)	6 (8.7%)	
Weekly income	No income	24 (7.7%)	57 (13.4%)	59 (10.6%)	40 (18.1%)	40 (17.7%)	11 (15.9%)	<0.001
	$1–$399	89 (28.6%)	114 (26.8%)	176 (31.7%)	71 (30.3%)	71 (31.4%)	30 (43.5%)	
	$400–$649	39 (12.5%)	66 (15.5%)	96 (17.3%)	28 (13.9%)	28 (12.4%)	7 (10.1%)	
	$650–$999	54 (17.4%)	59 (13.9%)	90 (16.2%)	34 (15.7%)	34 (15.0%)	6 (8.7%)	
	$1000–$1499	46 (14.8%)	63 (14.8%)	46 (8.3%)	23 (10.0%)	23 (10.2%)	7 (10.1%)	
	$1500–over $3000	47 (15.1%)	45 (10.6%)	47 (8.5%)	14 (4.4%)	14 (6.2%)	3 (4.3%)	
	Prefer not to say	12 (3.9%)	21 (4.9%)	42 (7.6%)	16 (7.6%)	16 (7.1%)	5 (7.2%)	

^1^ One-way ANOVA with post-hoc testing for Age and BMI. Tests were adjusted for all pairwise comparisons using the Bonferroni correction. Pearson’s chi-square performed for Gender, studying, and Income. Values in rows not sharing the same subscript are significantly different from one another; ^2^ Mean (Standard Deviation), all such values; ^3^
*n* (%), all such values.

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
