# Peer review of "Beyond Body Weight: Design and Validation of Psycho-Behavioural Living and Eating for Health Segments (LEHS) Profiles for Social Marketing"

_nutrients, 2020, doi:10.3390/nu12092882_

Round 1

Reviewer 1 Report

I can see the value of developing this profiling tool and its strength in designing appropriate interventions to tackle obesity/improve obesity related behaviours. I believe the application of social marketing techniques is a very promising approach, especially in this age group, and this study has obviously followed a thorough and considered process. I found the validity descriptions particularly useful. However, I feel that the article needs some reframing and clearer reporting of the development of LEHS. The relevance of some of the study processes were not clear to me. I also feel that the addition of information on the practical use of the LEHS profiles would increase the usefulness of this research for real-world practice.

I have three major concerns with the article, followed by some further minor considerations:

  1. My first main issue with the article is the argument of the use of BMI in informing intervention development. I agree with the authors that this is not necessarily a useful basis for intervention development and profiling of target participants; however, the authors use a relatively large section of the introduction critiquing BMI in terms of its clinical diagnostic capabilities and I do not feel this of relevance to the main argument for the use of psycho-behavioural profiling, which, as I understand, the authors are not claiming to be used as a clinical diagnostic measure.
  2. My second concern is some lack of relevance, clarity and detail on the development of the LEHS profiles. Firstly, it is unclear how the literature reviews inform the LEHS development? They seem to just focus on the acceptability of social media use within interventions. Whilst this provides good justification for the development of LEHS, I do not see how that information gathered maps on to the LEHS descriptions. These review findings seem to be more suited as part the introduction unless a clearer link to each profile development is given. This also links to my second point, the description of the development of the LEHS profiles and the thematic analysis is quite sparse for what is a major stage of the process. The authors refer to ‘thematic analysis of the data…’ but some clarity on what these data are in particular and the coding process followed is required. Is this mainly data from the online conversations, or did the formative research stage inform the analysis equally? Was analysis of each data source conducted simultaneously or focused on one data source with clarification from others? It is also unclear if the profiles were finalised before the survey was conducted or if further refinement took place as a result of the findings from this final stage.
  3. Finally, I feel that the article would be strengthened with some practical examples of intervention strategies that are applicable to each of the profiles developed. I believe this would increase the potential impact and practical translation of the research, and support the main aim of the profile development. Or if the authors feel the identification of suitable strategies requires further investigation, some discussion around this and how this might conducted is needed. As it stands, it feels like the article is missing this final part of the jigsaw.

Minor considerations:

  1. I agree that BMI not a useful measure for differentiating between profiles but do think the BMI of each group provides some interesting/useful data, as does the income/education data. It is important to understand if there are differences in psycho-behavioural profiles across demographic groups that could potentially explain inequalities in health, or be used to prevent further widening, and it would be good to see some discussion around this. It is, of course, difficult to determine SEP in this age group using the proxy measures of education and income. For future work, indication of parent SEP would be useful.
  2. The authors raise an important point about wider reasons for being ‘unconcerned about being healthy’ but I think this also highlights a potential limitation in the sensitivity of the profiles developed. Being unable to financially afford to eat healthily does not fully fit with the Blissfully Unconcerned profile. These individuals may of course be very ‘concerned’ about being healthy and therefore this profile may not be able to distinguish between attitude and ability. As it is currently written the discussion around food insecurity seems disjointed (a bit of a jump).
  3. Consideration of the cost effectiveness of social media interventions using tailoring to psycho-behavioural profiles would be useful (it seems that it could be a useful and cost effective approach). Again, this could strengthen the impact and practical translation.

Author Response

Response to Reviewer 1’s comments

Point 1: I can see the value of developing this profiling tool and its strength in designing appropriate interventions to tackle obesity/improve obesity related behaviours. I believe the application of social marketing techniques is a very promising approach, especially in this age group, and this study has obviously followed a thorough and considered process. I found the validity descriptions particularly useful. However, I feel that the article needs some reframing and clearer reporting of the development of LEHS. The relevance of some of the study processes were not clear to me. I also feel that the addition of information on the practical use of the LEHS profiles would increase the usefulness of this research for real-world practice.

Response 1: Thank you for reviewing our manuscript and for your suggestions to help improve the reporting of the study.

Point 2: I have three major concerns with the article, followed by some further minor considerations: My first main issue with the article is the argument of the use of BMI in informing intervention development. I agree with the authors that this is not necessarily a useful basis for intervention development and profiling of target participants; however, the authors use a relatively large section of the introduction critiquing BMI in terms of its clinical diagnostic capabilities and I do not feel this of relevance to the main argument for the use of psycho-behavioural profiling, which, as I understand, the authors are not claiming to be used as a clinical diagnostic measure.

Response 2: We have reduced this paragraph and removed some of the information (Lines 62-68 on Track Changes version of revised manuscript) regarding BMI as a clinical diagnostic.

Point 3: My second concern is some lack of relevance, clarity and detail on the development of the LEHS profiles. Firstly, it is unclear how the literature reviews inform the LEHS development? They seem to just focus on the acceptability of social media use within interventions. Whilst this provides good justification for the development of LEHS, I do not see how that information gathered maps on to the LEHS descriptions. These review findings seem to be more suited as part the introduction unless a clearer link to each profile development is given. This also links to my second point, the description of the development of the LEHS profiles and the thematic analysis is quite sparse for what is a major stage of the process. The authors refer to ‘thematic analysis of the data…’ but some clarity on what these data are in particular and the coding process followed is required. Is this mainly data from the online conversations, or did the formative research stage inform the analysis equally? Was analysis of each data source conducted simultaneously or focused on one data source with clarification from others? 

Response 3: Thank you for this suggestion, we agree our reporting of the LEHS development and contribution of the literature reviews and formative research in the development of these could be more clear. We have added the following to clarify our position:

“The literature reviews conducted prior to the qualitative analysis of the online conversations shaped the context of the food and health environment in which the young adults in the online conversations were living and this was used as a lens while analysing.” (Lines 214-216)

“The formative research from the online conversations was conducted additionally to the LEHS thematic analysis in order to explore in depth specific topics within the online conversations. This formative research helped to inform and validate the LEHS profiles.” (Lines 232-234)

“Draft profiles were developed based on thematic analysis of the online conversations data and background literature reviews (K.K., L.B., M.K.). Qualitative data analysis used two different approaches: hypothesis-generating (to uncover new themes not previously identified by literature reviews) and hypothesis-testing to identify statements relating to the Integrated Model of Behaviour Change (IMBC) model [46] . For the hypothesis-generating analysis, the online conversations were analysed in an exploratory way, using a hand-coding process on paper and using NVivo qualitative data analysis Software; QSR International Pty Ltd. Version 11. Researchers used thematic analysis to examine the qualitative data collected from participants’ discussions and identify themes using a constant comparison approach [47].  Common themes associated with healthy eating behaviour, using the IBMC model components were drawn out and used as a basis for considering likely LEHS profiles. Additionally, comparative analysis will be used on a continual basis to compare people and each group to check that LEHS profiles are appropriately represented by the data. Investigator triangulation enhanced the transferability and dependability of research findings [48].” (Lines 330-342)

Point 4: It is also unclear if the profiles were finalised before the survey was conducted or if further refinement took place as a result of the findings from this final stage.

Response 4: On lines 387-388, we have added a sentence that clarifies that the LEHS profiles were finalised before the survey was conducted.

“Following multiple iterations and development of the profiles on a feedback cycle by the research team members, an expert panel, and Think Tank, the outcome LEHS profiles were then quantitatively defined using an online survey methodology.”

Point 5: Finally, I feel that the article would be strengthened with some practical examples of intervention strategies that are applicable to each of the profiles developed. I believe this would increase the potential impact and practical translation of the research, and support the main aim of the profile development. Or if the authors feel the identification of suitable strategies requires further investigation, some discussion around this and how this might conducted is needed. As it stands, it feels like the article is missing this final part of the jigsaw.

Response 5: Thank you for highlighting this gap. Social marketing strategies targeting LEHS will be co-designed and co-created by young adults (18-24 year olds) as part of the next phase of our research. This is outlined in our protocol paper. We have addressed your comment by clarifying this on line 513-516.

“As outlined in Communicating Health protocol paper, the next phase of our research involves collaborating with young adults (18-24 year olds) in co-designing and co-creating strategies that target each LEHS [22]. This phase is currently ongoing and in data collection phase.”

Point 6: Minor considerations:

I agree that BMI not a useful measure for differentiating between profiles but do think the BMI of each group provides some interesting/useful data, as does the income/education data. It is important to understand if there are differences in psycho-behavioural profiles across demographic groups that could potentially explain inequalities in health, or be used to prevent further widening, and it would be good to see some discussion around this. 

Response 6: We agree that the differences in demographics between the LEHS is of importance and potentially contributes to their differing attitudes and values that distinguish the different LEHS. We have added a sentence relating to this on lines 457-460. 

“These differences in demographics between groups may have shaped the differing attitudes and values that distinguish the LEHS and adds to the defining qualities of the profiles. We encourage continued collection of such data alongside the use of the LEHS as we recognise that multiple factors determine attitudes and values.”

Point 7: It is, of course, difficult to determine SEP in this age group using the proxy measures of education and income. For future work, indication of parent SEP would be useful.

Response 7: Based on the data we collected we were not able to measure socioeconomic position but agree it would be useful for future work. We have added this to the limitations and future directions.

“The data collected did not allow for measurement of socio-economic position beyond income and education status and therefore future research would benefit from the inclusion of a measure of socio-economic status.” (Lines 547-549)

Point 8: The authors raise an important point about wider reasons for being ‘unconcerned about being healthy’ but I think this also highlights a potential limitation in the sensitivity of the profiles developed. Being unable to financially afford to eat healthily does not fully fit with the Blissfully Unconcerned profile. These individuals may of course be very ‘concerned’ about being healthy and therefore this profile may not be able to distinguish between attitude and ability. As it is currently written the discussion around food insecurity seems disjointed (a bit of a jump).

Response 8: We have added some clarification around the potential link between the lower income observed in this group and their attitude towards eating and have also removed the discussion on food security.

“The ‘Blissfully Unconcerned’ had a high percentage of people with low incomes which may indicate be indicative of food security issues whereby they possibly cannot afford to be concerned about eating healthily. They may have other concerns that are more pertinent than what they eat or not believe they have the choice financially to consume healthy foods, therefore potentially affecting their overall indifferent attitude towards healthy eating.” (Lines 461-465)

Point 9: Consideration of the cost effectiveness of social media interventions using tailoring to psycho-behavioural profiles would be useful (it seems that it could be a useful and cost effective approach). Again, this could strengthen the impact and practical translation.

Response 9: Thank you for this input, we agree that this is a benefit of using social media strategies. The focus of this paper is on the development of LEHS and as mentioned in Response 5, social media strategy development is the next phase of our research and currently still ongoing in the data collection phase. We will make a note regarding your comment here to include in our next publication.

Reviewer 2 Report

This is an interesting, carefully researched and nicely written paper. I also thought the diagrams and tables added value to the paper. Table 3 was especially of interest.

Only a very few comments:

  • In paragraph starting on line 86, I think it might be worth just adding one sentence stating that it is about moving beyond simply segmenting by demographics and geography as is done in a lot of public health projects aiming to tackle obesity. I find a lot of public health professionals say ‘well, I segment already’, but they don’t segment by psychographics or current behaviour, as we do in social marketing. That is why having a sentence up front showing how different might be helpful to readers from a non-social marketing background here. You do mention it on line 405 but I think needs to be up-front also.
  • I had to read the paragraph starting on line 131 a couple of times before I fully understood it all. For ease of reading it, it might be helpful to illustrate what you are meaning with some examples:

(you say on line 133) In most studies, validation efforts often focus on achieving a valid ‘measurement’. For example xxx

I think this would help the reader as I was left thinking – I think they mean x, but I am not sure.

  • I may have missed it, but was perception of risk in relation to obesity explored as a variable? With the obesity work I do in the middle eat, this is a key factor (there is no or very little perception of risk).

There were a couple of points I was left wondering after reading the paper and which maybe should be added to the discussion.

  • On line 362 you state that eating behaviours are highly complex. I think this point is very important that needs elaborating further. I think you need to state somewhere in your discussion how eating habits and tastes are both personal and entwined in our cultural, religious, and nationalistic norms. This means that people do not make decisions around food choices and nutrition in isolation or based solely on health outcomes/perceived health benefits. So, would ethnicity/cultural background make a difference? I think Australia has a large Greek population (?), do factors such as cultural heritage impact then on the segments? I was left pondering that question. Or if not known, then that is an area for further research or is maybe a limitation? Are there any other limitations you need to highlight to the readers?
  • Finally, I wondered – how applicable are the findings outside of Australia? As this journal is read widely others might think the same and wonder what the authors think.

Author Response

Response to Reviewer 2’s comments

Point 1: This is an interesting, carefully researched and nicely written paper. I also thought the diagrams and tables added value to the paper. Table 3 was especially of interest.

Response 1: Thank you for reviewing our manuscript and for your suggestions to help improve the reporting of the study.

Point 2:

Only a very few comments:

In paragraph starting on line 86, I think it might be worth just adding one sentence stating that it is about moving beyond simply segmenting by demographics and geography as is done in a lot of public health projects aiming to tackle obesity. I find a lot of public health professionals say ‘well, I segment already’, but they don’t segment by psychographics or current behaviour, as we do in social marketing. That is why having a sentence up front showing how different might be helpful to readers from a non-social marketing background here. You do mention it on line 405 but I think needs to be up-front also.

Response 2: You have made a good point here, thank you. We have added your suggestion on line 90-95 (on Track Changes version of revised manuscript).

“These methods are often combined with demographic and geographic methods to produce a nuanced profile of the targeted group. Segmentation by demographics (such as age and gender) and geography are common methods used in the public health sector, but can be quite limiting in terms of developing an in-depth understanding of the target group [17]. Therefore, the Communicating Health project uses a psycho-behavioural approach to segmentation that comprises a combination of methods of segmentation [29]”

Point 3: I had to read the paragraph starting on line 131 a couple of times before I fully understood it all. For ease of reading it, it might be helpful to illustrate what you are meaning with some examples:

(you say on line 133) In most studies, validation efforts often focus on achieving a valid ‘measurement’. For example xxx I think this would help the reader as I was left thinking – I think they mean x, but I am not sure.

Response 3: We have simplified the sentence - line 137 and added examples referring to ‘measurement’ (line 139-141).

“In most studies, validation efforts often focus on achieving a valid ‘measurement’ (e.g. constructs that are reliable and reproducible).”

Point 4: I may have missed it, but was perception of risk in relation to obesity explored as a variable? With the obesity work I do in the middle eat, this is a key factor (there is no or very little perception of risk).

Response 4: In the online conversations we did explore the risk perception of obesity however through thematic analysis this was not a defining characteristic of the different LEHS and therefore was not widely discussed in this paper. 

Point 5: There were a couple of points I was left wondering after reading the paper and which maybe should be added to the discussion.

  • On line 362 you state that eating behaviours are highly complex. I think this point is very important that needs elaborating further. I think you need to state somewhere in your discussion how eating habits and tastes are both personal and entwined in our cultural, religious, and nationalistic norms. This means that people do not make decisions around food choices and nutrition in isolation or based solely on health outcomes/perceived health benefits. So, would ethnicity/cultural background make a difference? I think Australia has a large Greek population (?), do factors such as cultural heritage impact then on the segments? I was left pondering that question. Or if not known, then that is an area for further research or is maybe a limitation? Are there any other limitations you need to highlight to the readers?

Response 5: We agree that it would be good to elaborate on the complexity of food choice and as we didn’t specifically look at the effect of cultural heritage on the LEHS profiles have added this into the limitations and directions for future research section (lines 451-455). We have also included here some additional limitations of the research.

“The complexity of eating behaviours and the interplay between culture, taste preference, religion and many other factors makes food choices highly individualistic. People do not make food choices based on just one factor such as the health outcomes of food. In this study we did not look at how religion and culture shape the designation into different LEHS groups and this warrants investigation in the future.”

Point 6: Finally, I wondered – how applicable are the findings outside of Australia? As this journal is read widely others might think the same and wonder what the authors think.

Response 6: We have added in clarification that these segments would need to be tested in other populations as it is not clear whether they would be applicable outside the Australian young adult population in which they were created and tested.

“As these LEHS profiles were based on and tested in a sample of Australian young adults, it is unclear whether the same profiles would exist in different population groups. Further use of the profiles in different population would therefore require validation to ascertain the applicability of these LEHS profiles.” (lines 455-458)

Reviewer 3 Report

Dear authors;   I have no major changes, just that the authors clarify in a more insightful way the conclusions they arrive after reviewing the literature and also if they are able to include other fields of knowledge where the findings can be applied, for example happiness management.

After reviewing the paper, I want to point out that the paper is well written and it is easy to read. Moreover, the use of the dietary habits-social marketing construct is very original in the actual society youth area. Besides, the research results are evidence-based and the conclusions are consistent with the arguments presented.   

Author Response

Response to Reviewer 3’s comments

Point 1: Dear authors;   I have no major changes, just that the authors clarify in a more insightful way the conclusions they arrive after reviewing the literature and also if they are able to include other fields of knowledge where the findings can be applied, for example happiness management.

Response 1: Thank you for your input, we have added how these LEHS would benefit other fields of knowledge.

“Furthermore, we believe these LEHS would also benefit other fields, in particular when examining other aspects of living a healthy lifestyle such as fitness/physical exercise and mental health management.” (Lines 448-450 on Track Changes version of revised manuscript)

Point 2: After reviewing the paper, I want to point out that the paper is well written and it is easy to read. Moreover, the use of the dietary habits-social marketing construct is very original in the actual society youth area. Besides, the research results are evidence-based and the conclusions are consistent with the arguments presented.   

Response 2: Thank you for reviewing our manuscript and for your suggestions to help improve the reporting of the study.

Round 2

Reviewer 1 Report

Thank you for addressing each of my comments. I am satisfied that the revisions that you have made have addressed these and I am happy to approve the article for publication.